# A Robust Training Method for Federated Learning with Partial Participation

## Abstract

Client weighting and partial participation are key techniques in federated learning. They reduce communication costs and maintain a balance in the data used for model training. Numerous strategies are well-established within the research community, leading to growing interest in developing a unified theory. In this paper, we explore this issue in detail. We propose a method that accumulates unused gradients from the current iteration locally and, after full aggregation, leverages them for effective training. Our framework supports a wide class of weighting and sampling heuristics. Furthermore, we show the proposed approach to be robust against clients' periodic disconnection. To validate it, we conduct a series of numerical experiments involving the training of convolutional and transformer-based architectures.

## 1 Introduction

Optimization is a cornerstone of training machine learning and neural network models. In a nutshell, almost every AI-based solution aims to minimize an empirical risk [Shalev-Shwartz et al., 2010], which evaluates how well the data is approximated. This process involves adjusting parameters to reduce the discrepancy between predicted outputs and ground truth labels, thereby improving generalization performance. Formally, the problem can be expressed as

$$\min_{x \in \mathbb{R}^d} \left[ \frac{1}{n} \sum_{i=1}^{n} \ell(g(x, a_i), b_i) \right], \tag{1}$$

where $x$ denotes the trainable parameters of the model $g$, $(a_i, b_i)$ is the $i$-th sample from the dataset with size $n$, and $\ell$ is the loss function. Nowadays, there is a variety of methods developed to efficiently solve (1) [Robbins and Monro, 1951, Nesterov, 1983, Kingma and Ba, 2014, Defazio and Mishchenko, 2023]. The current successes of machine/deep learning owe much to the development of powerful numerical techniques that enable training on a huge amount of samples. Large-scale data processing became possible with the advancement of distributed optimization [Verbraeken et al., 2020]. Instead of solving the problem on a single machine, samples are shared among $M$ nodes/devices/clients/machines connected via a server. Hence, the problem (1) transforms into

$$\min_{x \in \mathbb{R}^d} \left[ f(x) = \frac{1}{M} \sum_{m=1}^{M} f_m(x) = \frac{1}{M} \sum_{m=1}^{M} \frac{1}{n_m} \sum_{i_m=1}^{n_m} \ell(g(x, a_{i_m}), b_{i_m}) \right], \tag{2}$$

where $n_m$ is the size of the dataset, stored on $m$-th device.

### 1.1 Client Weighting

Parallel data processing helps to reduce computational time significantly [Zinkevich et al., 2010, Abadi et al., 2016, Jouppi et al., 2017]. However, contemporary applications present new challenges.

Submitted to 39th Conference on Neural Information Processing Systems (NeurIPS 2025). Do not distribute.

Training samples are often accumulated locally by each specific machine, rather than being collected and distributed manually. This paradigm with data remaining on edge devices is called federated learning [Konečnỳ et al., 2016, McMahan et al., 2017, Bonawitz et al., 2019]. In such a setup, local datasets are typically heterogeneous – they vary in size, distribution, and quality. For instance, one device may hold unique objects that are poorly represented across the rest of the network, but are crucial for capturing more dependencies. This leads to the conclusion that some clients may be more useful than others. Modern approaches usually assign dynamic weights $\{\pi_m\}_{m=1}^M$ and use

$$f(x) = \sum_{m=1}^M \pi_m f_m(x), \ \ \text{s.t.} \ \pi_m > 0, \sum_{m=1}^M \pi_m = 1 \tag{3}$$

to calculate statistics. If the devices are considered to be equivalent, this corresponds to the case where $\pi_1 = \ldots = \pi_M = 1/M$. As a result, more important nodes contribute more significantly to the global loss. There are many strategies to prioritize the clients known in the literature.

**Weighting Based on Data Quality/Quantity.** The most straightforward way to cope with data imbalance is to consider a number of local samples. McMahan et al. [2017] suggested setting each coefficient as the constant $\pi_m = n_m/n$. Since then, many modifications of this approach have been proposed, including federated averaging schemes with momentum [Wang et al., 2019, Reddi et al., 2020], variance reduction [Liang et al., 2019, Karimireddy et al., 2020] and proximal updates [Li et al., 2020]. However, this type of weighting ignores heterogeneity in terms of data quality, leading to bias, e.g. if some client holds an enormous amount of objects with the same labels. To support the diversity of training samples, Yurochkin et al. [2019] proposed to match the neurons of client neural networks before averaging. Building on the foundations laid by this work, subsequent works have explored more efficient approaches extensively [Wang et al., 2020a, Zhang et al., 2022, Yang et al., 2023, Wu et al., 2023, Kafshgari et al., 2023].

**Learned Weighting Strategies.** It is also common to learn weighting strategies instead of using fixed heuristics. Mohri et al. [2019] were among the first to present results in this direction. They proposed solving the saddle-point problem $\min_{x \in \mathbb{R}^d} \max_{\pi \in \triangle_1^M} \sum_{m=1}^M \pi_m f_m(x)$ to give small weights to well-trained devices. The idea of optimizing agnostic empirical loss was then generalized by Li et al. [2019a]. Their `q-FedAvg` can be reduced to agnostic optimization as one of the special cases. However, in practice, it is hard to search for appropriate saddle-points [Daskalakis and Panageas, 2018, Jin et al., 2020], especially in federated learning [Sharma et al., 2023]. As a result, the community has shifted towards softer adaptive approaches based on local losses [Zhang et al., 2020, Gao et al., 2022] and gradients [Wang et al., 2020b, Luo et al., 2024].

**Robust Weighting.** The idea of assigning weights to the devices found its application in robust optimization, where malicious clients can disrupt the learning process [Baruch et al., 2019, Xie et al., 2020, Fang et al., 2020]. To combat such attacks, advanced schemes usually compute $\{\pi_m\}_{m=1}^M$, as the trust scores of the devices based on their objectives decrease [Xie et al., 2019], local gradients [Cao et al., 2020, Yan et al., 2023], and the number of local samples [Cao and Lai, 2019]. Recently, researchers came up with the idea of using a Bayesian approach [Yang et al., 2024].

## 1.2 Client Sampling

Another significant issue of federated learning, on par with heterogeneity, is the communication bottleneck [Tang et al., 2020, Shi et al., 2020]. Sharing information between machines is costly and can limit the positive effect of parallelism, which is especially tangible when clients send messages to the server [Kairouz et al., 2021]. This issue is magnified in federated learning, where edge devices may have unstable network connectivity, and transmitting large updates may be prohibitively slow. Many techniques exist to reduce communication [Seide et al., 2014, Alistarh et al., 2017, Stich, 2018]. Partial participation is a special one among them [Li et al., 2019b, Yang et al., 2021]. In each communication round, only a random subset of clients participates in training, while the rest remain inactive. This approach offloads the server by decreasing the number of updates that need to be aggregated. Moreover, it provides significant advantages in edge computing, where communication channels are not equivalent, or some of them may be unavailable. Nowadays, there is a wide range of heuristics, which allows to choose subset of clients efficiently.

**Data-Based Sampling Strategies.** Methods from this class rely on zero- and first-order information of local functions. `Importance Sampling FedAvg` [Rizk et al., 2021] was one of the first such approaches. The authors suggested evaluating the relevance of a device by how large its gradient is relative to the others. Indeed, a small gradient makes a weak contribution to the step. Consequently, communication with this node can be neglected. Nguyen et al. [2020] proposed an orthogonal approach. Their `FOLB` measures the angle between local and average gradient. If it is negative, then such a device is useless at the current moment. This idea was then developed extensively in [Wu and Wang, 2022, Zhou et al., 2022]. In addition, techniques based on the norms of updates [Chen et al., 2020] and local loss decrease [Cho et al., 2022] were proposed. There are also a number of approaches that dynamically exploit data heterogeneity to maintain balance [Zhang et al., 2023] or support diversity [Chen and Vikalo, 2024].

**System-Based Sampling Strategies.** Another approach is to use information about the network itself. `FedCS` [Nishio and Yonetani, 2019] categorizes clients into groups based on their computational power. This strategy saves wall-clock time by avoiding frequent selection of weak devices. Another class of techniques optimizes energy consumption [Xu and Wang, 2020]. Most modern system heterogeneity techniques also incorporate local data considerations [Lai et al., 2021, Li et al., 2022]. `F3AST` [Ribero et al., 2022] learns an availability-dependent client selection strategy to minimize the impact of variance on the global model's convergence.

Thus, the community came up with various techniques for weighting and sampling to make partial participation as efficient as possible. The development of each new scheme was challenging in terms of algorithm design and convergence proof. Consequently, a number of papers appeared attempting to propose a theory without utilizing the properties of any particular strategy.

### 1.3 Unification of Sampling Strategies

Existing papers in this area of research are built around the federated averaging scheme [McMahan et al., 2017]. Li et al. [2019b] proposed an analysis for strongly convex objectives, obtaining a sublinear convergence rate $\mathcal{O}\left(\kappa^2/K\right)$, where $\kappa$ is the condition number. However, they modeled the partial participation environment via unbiased sampling. Cho et al. [2022] were the first to study the unified case with biased devices selection. They derived $\mathcal{O}\left(\kappa^2/K + \kappa Q\right)$, where $Q$ is a non-vanishing term that becomes zero solely in the absence of sampling bias. Thus, the authors recovered the results of Li et al. [2019b], but failed to extend the theory to weaker assumptions. The first success in this direction was achieved in [Luo et al., 2022]. This work resolved key questions regarding biased sampling in the strongly convex case. However, the non-convex analysis holds greater significance for applications. For this setting, Wang and Ji [2022] obtained $\mathcal{O}\left(\sqrt{L}/\sqrt{K} + \delta\right)$, where $L$ is the smoothness constant and $\delta$ is the uniform bound on the difference between local gradients. This result contains the non-vanishing term and does not match the lower bound $\Omega\left(L/K\right)$ [Carmon et al., 2020].

Thus, current works in this field rely on `FedAvg`. As a consequence, their analysis requires boundedness of gradients [Li et al., 2019b, Cho et al., 2022, Luo et al., 2022] or their differences [Wang and Ji, 2022] even in the non-stochastic case. Therefore, there is still no flawless unified theory of partial participation.

### 1.4 Our Contribution

In contrast to prior works, where partial participation analysis was built upon `FedAvg`, we introduce our own scheme to leverage client sampling. While existing techniques ignore the information from inactive clients, our approach utilizes it for benefits. Namely, devices accumulate gradient surrogates locally, and the server accounts for them after the full aggregation round. The proposed approach allows weighting and sampling clients according to a variety of strategies, including biased ones. The convergence of our scheme can be proven in both strongly convex and non-convex cases without introducing unnatural assumptions. The obtained rates do not contain non-vanishing terms. To validate the theory, we conduct experiments with RESNET-18 and VIT.

## 2 Setup

We begin presenting our results with assumptions necessary to prove convergence. First of all, the objective is assumed to be smooth. This requirement is well-established in optimization.

**Assumption 1.** *The function $f$ is L-smooth, i.e. for all $x, y \in \mathbb{R}^d$ it satisfies*

$$\|\nabla f(x) - \nabla f(y)\| \leqslant L\|x - y\|.$$

Neural networks tend to have a complex loss landscape [Cybenko, 1989, Nguyen and Hein, 2018]. Since we are motivated by real-world scenarios, our main goal is to prove convergence in the non-convex case. For completeness, we also derive results under stronger assumptions.

**Assumption 2.** *The function $f$ is:*

*(a) **non-convex** with at least one global minimum:*

$$\text{there exists may be not unique,} \ \ x^* \ \ s.t. \ \ f(x^*) = \inf_{x \in \mathbb{R}^d} f(x) > -\infty.$$

*(b) $\mu$-**strongly convex**, i.e. for all $x, y \in \mathbb{R}^d$ it satisfies*

$$f(y) \geqslant f(x) + \langle \nabla f(x), y - x \rangle + \frac{\mu}{2}\|y - x\|^2.$$

Federated learning methods usually require a bound on data heterogeneity to provide convergence guarantees [Khaled et al., 2020, Karimireddy et al., 2020]. In our work, we quantify it via gradients [Tang et al., 2018, Stich, 2020].

**Assumption 3.** *Each gradient $\nabla f_m$ is similar to the full gradient $\nabla f$, i.e. for all $x \in \mathbb{R}^d$ it satisfies*

$$\frac{1}{M}\sum_{m=1}^{M}\|\nabla f_m(x) - \nabla f(x)\|^2 \leqslant \delta_1\|\nabla f(x)\|^2 + \delta_2.$$

This assumption is not too strict, since we do not require uniform boundedness ($\delta_1 = 0$). The following one is imposed to derive convergence of our algorithm with local stochasticity. If one removes it, our theory still holds.

**Assumption 4.** *Each worker has access to a stochastic gradient $\nabla f_m(x, \xi_m)$. This is an unbiased random variable with bounded variance, i.e. for all $x \in \mathbb{R}^d$ it satisfies*

$$\mathbb{E}_{\xi_m}\left[\nabla f_m(x, \xi_m)\right] = \nabla f_m(x),$$
$$\mathbb{E}_{\xi_m}\left[\|\nabla f_m(x, \xi_m) - \nabla f_m(x)\|^2\right] \leqslant \sigma^2.$$

This assumption appears in different forms in a number of classic papers [Stich, 2018, Gower et al., 2019, Gorbunov et al., 2020]. Next, we consider that weights $\{\pi_m\}_{m=1}^{M}$ from (3) lie on the regularized simplex. Namely, $\pi \in \Delta_1^M \cap \left(\bigcap_{m=1}^{M}\left\{\pi : e_m^\top \pi + \frac{\alpha}{M} \geqslant 0\right\}\right)$, where $1 \leqslant \alpha \leqslant M$ is the regularization parameter and $e$ is the unit basis. This technique is useful for solving a wide range of tasks [Mehta et al., 2024].

## 3 Algorithms and Analysis

### 3.1 Motivation

Existing papers on the unification of client sampling consider `FedAvg` without any modifications. Section 1.3 suggests that this approach is not promising due to poor results even under strong assumptions. A potential direction for future research could be to find a more suitable scheme. Below we propose an intuition that helps to address this issue.

To understand biased sampling, Cho et al. [2022] introduced the definition of selection skew and utilized it in the analysis. This is exactly the cause of the non-vanishing term in their rate. Indeed, there is no convergence if, for example, some devices are never selected for communication. However, we propose that the problem could be solved if we could somehow account for the error accumulated due to bias. To develop this idea, we formalize the sampling strategy as follows. First, we assign weights $\pi_m$ to devices, as described in (3). Next, we define the selection rule of the server as a stochastic operator $\mathcal{R} : \mathbb{R}^M \to \mathbb{R}^M$ that zeros some entries of the input vector while retaining the others. Applying this operator to the introduced vector of weights, it can be seen that the wide variety of strategies described in Section 1.2 fits this formalism. This applies not only to simple cases of selecting clients with the highest weights but also to non-trivial ones, such as zeroing the weights of unavailable nodes.

Viewing partial participation as weight vector sparsification reveals connections to well-studied techniques [Beznosikov et al., 2023]. A state-of-the-art technique to handle it efficiently is error

feedback [Stich and Karimireddy, 2020, Richtárik et al., 2021]. Since sampling rules are represented as compressors, we believe that this idea may be extremely useful in our setting as well. However, we cannot apply the existing framework directly, as it requires all clients to account for the error at each algorithm iteration. Error feedback was designed to compress the information, while our goal is to exclude some clients from an entire epoch.

Thus, we have to address this challenge before proceeding to a unified analysis of partial participation.

### 3.2 Partial Participation without Unavailable Devices

To develop the idea proposed in Section 3.1, we present the **P**artial **P**articipation with **B**ias **C**orrection framework (PPBC, see Algorithm 1) that supports a wide class of weighting and sampling approaches. Since computing full-batch gradients is often impractical in modern applications, we also account for local stochasticity.

---

**Algorithm 1** PPBC

---

1: **Input:** Start point $x^{-1,H^{-1}} \in \mathbb{R}^d$, $g^{-1,H^{-1}} \in \mathbb{R}^d$, epochs number $K$, number of devices $M$
2: **Parameters:** Stepsize $\gamma > 0$, momentum $0 < \theta < 1$, regularization $1 \leqslant \alpha \leqslant M$
3: **for** epochs $k = 0, \ldots, K-1$ **do**
4:      Initialize $\pi^k$ // *Server weighs clients using any procedure*
5:      $\hat{\pi}^k = \widehat{\mathcal{R}}^k(\pi^k)$ // *Server selects clients to communicate through epoch using any rule $\hat{\mathcal{R}}$*
6:      $g_m^{k,0} = 0$ // *Each client initializes the gradient surrogate*
7:      $x^{k,0} = x^{k-1,H^{k-1}} - g^{k-1,H^{k-1}}$ // *Server initializes the initial point of the epoch*
8:      Generate $H^k \sim \text{Geom}(p)$ // *Server generates number of iterations of k-th epoch*
9:      **for** iterations $h = 0, \ldots, H^k - 1$ **do**
10:         $\widetilde{\pi}^{k,h} = \widetilde{\mathcal{R}}^{k,h}\left(\hat{\pi}^k\right)$ // *Server selects clients to communicate at the current round using rule $\widetilde{\mathcal{R}}$*
11:         **for** devices $m = 1 \ldots M$ in parallel **do**
12:             $g_m^{k,h+1} = g_m^{k,h} + (1-\theta)\left(\frac{1}{M} - \widetilde{\pi}_m^{k,h}\right)\nabla f_m(x^{k,h}, \xi_m^{k,h})$ // *Update the gradient surrogate*
13:         **end for**
14:         **for** each device $m : \widetilde{\pi}_m^{k,h} \neq 0$ **do**
15:             Send $\nabla f_m(x^{k,h}, \xi_m^{k,h})$ to the server
16:         **end for**
17:         $x^{k,h+1} = x^{k,h} - \gamma\left[(1-\theta)\sum_{m=1}^{M}\widetilde{\pi}_m^{k,h}\nabla f_m(x^{k,h}, \xi_m^{k,h}) + \theta g^{k-1,H^{k-1}}\right]$ // *Server updates parameters*
18:      **end for**
19:      **for** devices $m = 1 \ldots M$ in parallel **do**
20:         Send $g_m^{k,H^k}$ to the server
21:      **end for**
22:      $g^{k,H^k} = \sum_{m=1}^{M} g_m^{k,H^k}$ // *Server aggregates gradient surrogates*
23: **end for**

---

**Description of Algorithm 1.** In Algorithm 1, the weights $\pi^k = (\pi_1^k, \ldots, \pi_M^k)^\top$ are computed according to any of the mentioned strategies at the beginning of each epoch (Line 4). Next, the rule $\widehat{\mathcal{R}}$ is applied to determine the participating machines (Line 5). Its output $\hat{\pi}^k$ contains zeros at positions corresponding to nodes that are not chosen to communicate with the server. Note that $\widehat{\mathcal{R}}$ is not necessarily constant. There are no theoretical restrictions to change it during the execution. For example, one can vary the number of participating devices. We also allow additional client sampling at each iteration of the epoch by introducing a rule $\widetilde{\mathcal{R}}$ (Line 10). We propose to aggregate local gradient surrogates during the epoch (Line 12). To provide intuition beyond this update, we give a toy example where each $\pi_m$ is equal to $1/M$. In this way, all inactive devices collect their gradients, while all active ones retain the vector $g_m$ from the previous iteration. In the practical case with various weights, each device accounts for its deviation from the uniform distribution $\pi_u = \{1/M\}_{m=1}^M$. Next, we use the accumulated vectors during the following epoch (Line 17). To handle the magnitude imbalance between the gradient and its surrogate, we employ a smoothing scheme with a small parameter $\theta$.

193 **Analysis of Algorithm 1.**  We utilize virtual sequences to derive convergence rates of PPBC. The
194 idea is to introduce an additional vector

$$\widetilde{x}^{k,h} = x^{k,h} - \gamma \sum_{m=1}^{M} g_m^{k,h}$$

195 and use it to prove convergence. Substituting Lines 10, 17 in this definition, we obtain

$$\widetilde{x}^{k,h+1} = \widetilde{x}^{k,h} - \gamma \left[ (1-\theta)\frac{1}{M}\sum_{m=1}^{M}\nabla f_m(x^{k,h},\xi_m^{k,h}) + \theta g^{k-1,H^{k-1}} \right].$$

196 This is an important technique for our method, since the sequence $\widetilde{x}$ is updated with the average
197 of gradients from all devices, contrary to the original $x$. However, the virtual update also contains
198 a combination of accumulated gradients from the previous epoch. We emphasize that handling
199 $g^{k-1,H^{k-1}}$ is one of the main theoretical challenges we address. We set the epoch size $H^k$ as a
200 geometrically distributed random variable and provide the following lemma.

**Lemma 1.** *Suppose Assumptions 3, 4 hold. We consider the epoch size $H^k \sim Geom(p)$ and*
202 $1 \leqslant \alpha \leqslant M$. *Then for Algorithm 1 it implies*

$$\mathbb{E}_{H^k}\mathbb{E}_{\xi_m^{k,0}}\ldots\mathbb{E}_{\xi_m^{k,H^k-1}}\left\|g^{k,H^k}\right\|^2 \leqslant \frac{24(1-\theta)^2\alpha(\delta_1+1)}{p^2}\mathbb{E}_{H^k}\left\|\nabla f(x^{k,H^k})\right\|^2 + \frac{48(1-\theta)^2\alpha\delta_2}{p^2}$$
$$+ \frac{24(1-\theta)^2\alpha\sigma^2}{Mp^2}.$$

203 Assumption 4 is required only to handle local stochasticity. If the devices are able to compute exact
204 gradients, Lemma 1 holds with $\sigma = 0$. For the details, see Appendix D. As a result, we obtain the
205 convergence theorem.

**Theorem 1.** *Suppose Assumptions 1, 2(a), 3, 4 hold. Then for Algorithm 1 with $\theta \leqslant \frac{\gamma L p^2}{2}$ and*
207 $\gamma \leqslant \frac{p}{384L\alpha(\delta_1+1)}$ *it implies that*

$$\frac{1}{K}\sum_{k=0}^{K-1}\mathbb{E}\left\|\nabla f(x^{k,0})\right\|^2 \leqslant \frac{16\left(f(x^{0,0})-f(x^*)\right)}{\gamma K} + \frac{768\gamma L\alpha\delta_2}{p} + \frac{384\gamma^2 L^2\alpha\delta_2}{p^3}$$
$$+ \frac{400\gamma L\alpha\sigma^2}{Mp} + \frac{192\gamma^2 L^2\alpha\sigma^2}{Mp^3}.$$

208 The main obstacle in proving Theorem 1 is the terms $\|g^{k,H^k}\|^2$ and $\|g^{k-1,H^{k-1}}\|^2$ that appear in
209 the analysis. Using Lemma 1, they can be screwed to $\|\nabla f(x^{k,H^k})\|^2$ and $\|\nabla f(x^{k-1,H^{k-1}})\|^2$,
210 respectively. The first norm is easy to analyze. Classically, it serves as a convergence criterion.
211 Eliminating the second one turns out to be challenging. To cope with it, we incorporate the surrogate
212 into the starting point of the epoch (Line 7). For the details, see Appendix D.1. With such an estimate,
213 there is a technique to choose the stepsize $\gamma$ appropriately to obtain convergence [Stich, 2019].

**Corollary 1.** *Under conditions of Theorem 1 Algorithm 1 with fixed rules $\widehat{\mathcal{R}}^k \equiv \widetilde{\mathcal{R}}^{k,h} \equiv \mathcal{R}$ needs*

$$\mathcal{O}\left(M\frac{M}{C}\left(\frac{\Delta L\alpha\delta_1}{\varepsilon^2} + \frac{\Delta L\alpha\delta_2}{\varepsilon^4} + \frac{\Delta L\alpha\sigma^2}{M\varepsilon^4}\right)\right)$$

215 *number of devices communications to reach $\varepsilon$-accuracy, where $\varepsilon^2 = \frac{1}{K}\sum\limits_{k=0}^{K-1}\mathbb{E}\left\|\nabla f(x^{k,0})\right\|^2$, $\Delta =$*
216 $f(x^{0,0}) - f(x^*)$ *and $C$ is the number of devices participating in each epoch.*

217 We also consider varying sampling rules $\widehat{\mathcal{R}}^k$ and $\widetilde{\mathcal{R}}^{k,h}$ to study corollaries of Theorem 1, see
218 Appendix D.1 for the details. In our work, the analysis is extended to the strongly convex case.

**Theorem 2.** *Suppose Assumptions 1, 2(b), 3, 4 hold. Then for Algorithm 1 with $\theta \leqslant \frac{p\gamma\mu}{4}$ and*
220 $\gamma \leqslant \frac{p^2}{96L\alpha(\delta_1+1)}$ *it implies that*

$$\mathbb{E}\left\|x^{K,0}-x^*\right\|^2 \leqslant \left(1-\frac{\gamma\mu}{8}\right)^K\left\|x^{0,0}-x^*\right\|^2 + \frac{8\gamma\alpha}{\mu p^3}\left(144\delta_2 + \frac{74\sigma^2}{M}\right).$$

221 As well as for the non-convex objective, suitable $\gamma$ can be chosen in Theorem 2.

**Corollary 2.** *Under conditions of Theorem 2 Algorithm 1 with fixed rules $\widehat{\mathcal{R}}^{k,h} \equiv \widetilde{\mathcal{R}}^{k,h} \equiv \mathcal{R}$ needs*

$$\widetilde{\mathcal{O}}\left( M\left(\frac{M}{C}\right)^2 \left(\frac{L}{\mu}\alpha\delta_1 \log\left(\frac{1}{\varepsilon}\right) + \frac{M}{C}\frac{\alpha\delta_2}{\mu^2\varepsilon} + \frac{\alpha\sigma^2}{\mu^2 C\varepsilon}\right)\right)$$

*number of devices communications to reach $\varepsilon$-accuracy, where $\varepsilon^2 = \mathbb{E}\left\|x^{K,0} - x^*\right\|^2$ and $C$ is the number of devices participating in each epoch.*

### 3.3 Partial Participation with Unavailable Devices

The previous section addresses partial participation when all devices are available to communicate with the server. Indeed, in Algorithm 1 each node receives the current parameters at the end of the iteration, but does not send its gradient. This is motivated by the fact that forwarding a message from the client to the server is much more expensive than the other way around [Kairouz et al., 2021]. However, in practice, some devices can become inactive periodically [Li et al., 2019b, Yang et al., 2021]. Namely, these machines not only refrain from transmitting information but also do not perform local computations. In this section, we extend our theory to cover the case where the actual parameters are sent to only a fraction of the clients.

**Description of Algorithm 2.** In this section we present the part of Algorithm 2 (see Appendix A) that reflects key differences from Algorithm 1. To design it, we refuse using the biased sampling rule $\widetilde{\mathcal{R}}$ during the epoch. Instead, we simulate outage probability of the $m$-th device as a Bernoulli random variable $\eta_m^{k,h} \sim \text{Be}(q_m)$ [Chung, 2000] (Line 9). To describe client disconnection formally, $\eta_m^{k,h}$ is used to update the gradient surrogates (Line 11) and to perform the step (Line 16). Thus, in practice, it is not necessary for an inactive device to know the actual parameters. We also normalize the computed gradients by factors $\{q_m\}_{m=1}^{M}$ to balance their magnitudes.

---

9: Generate $\eta^{k,h}$

11: $g_m^{k,h+1} = g_m^{k,h} + (1-\theta)\frac{\eta_m^{k,h}}{q_m}\left(\frac{1}{M} - \hat{\pi}_m^{k,h}\right)\nabla f_m(x^{k,h}, \xi_m^{k,h})$

16: $x^{k,h+1} = x^{k,h} - \gamma\left[(1-\theta)\sum\limits_{m=1}^{M}\frac{\eta_m^{k,h}}{q_m}\hat{\pi}_m^{k,h}\nabla f_m(x^{k,h}, \xi_m^{k,h}) + \theta g^{k-1,H^{k-1}}\right]$

---

**Analysis of Algorithm 2.** We formulate the results for both non-convex and strongly-convex cases.

**Corollary 3.** *Suppose Assumptions 1, 2(a), 3, 4 hold. Algorithm 2 with fixed rules $\widehat{\mathcal{R}}^k \equiv \widetilde{\mathcal{R}}^{k,h} \equiv \mathcal{R}$ needs*

$$\mathcal{O}\left( M\frac{M}{C}\frac{1}{\min\limits_{1\leqslant m\leqslant M} q_m}\left(\frac{\Delta L\alpha\delta_1}{\varepsilon^2} + \frac{\Delta L\alpha\delta_2}{\varepsilon^4} + \frac{\Delta L\alpha\sigma^2}{\varepsilon^4}\right)\right)$$

*number of devices communications to reach $\varepsilon$-accuracy, where $\varepsilon^2 = \frac{1}{K}\sum\limits_{k=0}^{K-1}\mathbb{E}\left\|\nabla f(x^{k,0})\right\|^2$, $\Delta = f(x^{0,0}) - f(x^*)$ and $C$ is the number of devices participating in each epoch.*

**Corollary 4.** *Suppose Assumptions 1, 2(b), 3, 4 hold. Algorithm 2 with fixed rules $\widehat{\mathcal{R}}^k \equiv \widetilde{\mathcal{R}}^{k,h} \equiv \mathcal{R}$ needs*

$$\widetilde{\mathcal{O}}\left( M\left(\frac{M}{C}\right)^2 \frac{1}{\min\limits_{1\leqslant m\leqslant M} q_m}\left(\frac{L}{\mu}\alpha\delta_1 \log\left(\frac{1}{\varepsilon}\right) + \frac{M}{C}\frac{\alpha\delta_2}{\mu^2\varepsilon} + \frac{M}{C}\frac{\alpha\sigma^2}{\mu^2\varepsilon}\right)\right)$$

*number of devices communications to reach $\varepsilon$-accuracy, where $\varepsilon^2 = \mathbb{E}\left\|x^{K,0} - x^*\right\|^2$ and $C$ is the number of devices participating in each epoch.*

For more details, see Appendix E. Note that $\min_{1\leq m\leq M} q_m$ is a constant lying in the interval $(0,1]$. Thus, the rates of Algorithm 2 do not differ significantly from those for Algorithm 1. The only deterioration occurs in the variance term associated with local stochasticity. Thus, if each device has an access to its exact gradient, there is no asymptotical difference compared to Corollaries 1 and 2.

### 3.4 Discussion

We analyzed a wide class of sampling and weighting techniques and proposed algorithms for different network scenarios. Their rates asymptotically coincide with the optimal ones for SGD-like approaches

[Stich, 2019]. Due to considering biased strategies, we obtained an additional factor $M/C$. Again analogizing to compression, this multiplier signifies compression power. It is a well-known fact that there is no theoretical improvement for methods built upon error-feedback [Richtárik et al., 2021, Beznosikov et al., 2023]. However, we recover the convergence of SGD in the case of full participation. Comparing our non-convex rate regarding the main term $\mathcal{O}\left(1/\varepsilon^2\right)$ with prior works, we note that it surpasses that in [Wang and Ji, 2022] $\left(\mathcal{O}\left(1/\varepsilon^4 + \delta_2\right)\right)$ both asymptotically and by the absence of the non-vanishing term. Next, comparing strongly-convex rates $\left(\mathcal{O}\left(\kappa \log 1/\varepsilon\right)\right)$, we are superior to [Cho et al., 2022] $\left(\mathcal{O}\left(\kappa^2/\varepsilon + \kappa\delta_2\right)\right)$ and [Luo et al., 2022] $\left(\mathcal{O}\left(\kappa/\varepsilon\right)\right)$. Moreover, both of these works lack non-convex analysis. We highlight that we soften assumptions from all aforementioned works.

# 4  Experiments

To validate our theoretical findings, we conduct a systematic empirical study comparing three optimization approaches — FedAvg [Reddi et al., 2020], SCAFFOLD [Karimireddy et al., 2020], and PPBC (Algorithm 1) — each integrated with the same client sampling technique. Crucially, we maintain a fixed strategy across all methods, deliberately decoupling the sampling mechanism from algorithmic innovations to focus specifically on its interaction with different optimization approaches. Our experiments assess their relative performance under identical conditions, including model architectures, benchmark datasets, and hardware configurations. Firstly, we detail the experimental setup, including the neural network architectures, benchmark datasets, and computational hardware configurations employed in our analysis.

**Experimental Setup.**  We evaluate each sampling strategy under three distinct data distribution scenarios: (**distr-1**) homogeneous (*i.i.d.*), (**distr-2**) heterogeneous with different classes on each client, and (**distr-3**) strongly heterogeneous configurations with varying client-specific data quantities and class distributions. Our benchmark experiments employ image classification on CIFAR-10 [Krizhevsky et al., 2009] using a RESNET-18 architecture [Meng et al., 2019], establishing a controlled testbed for comparative analysis of Algorithm 1 across the sampling strategies. Comprehensive details regarding data partitioning, model architecture, and dataset specifications are provided in Appendix B.

**Client Selection Rule.**  Notably, not all strategies included in our comparative analysis inherently incorporate a client selection mechanism. To ensure a fair and consistent evaluation, we uniformly applied the following selection rule across all methods:

$$\widehat{\mathcal{R}}^k = \text{Top}_C\left(\pi^k\right),$$

where $\text{Top}_C$ denotes taking $C > 0$ clients with the highest weights $\pi^k$. Consequently, the remainder of our experiments will focus exclusively on the formulation and analysis of weight update rules, while treating the client selection process itself as a fixed component of the experimental framework.

**Loss-aware client sampling.**  Building upon previous work, Cho et al. [2022] introduced the POWER-OF-CHOICE (PoC) strategy, which employs a weighted client sampling mechanism based on local loss values. Formally, the weight update rule can be expressed as:

1. The server assigns to all clients the probabilities proportional to the data size fractions

$$p_m = \frac{n_m}{\left(\sum_{m'=1}^{M} n_{m'}\right)}.$$

2. The global model is sent by the server to the selected $C$ clients, which compute and return their local loss values based on their datasets. Subsequently, the weights are updated:

$$\pi^k = \left(\left[\frac{1}{n_m}\sum_{i_m=1}^{n_m} \ell(g(x, a_{i_m}), b_{i_m})\right]\right)_{m=1}^{M}.$$

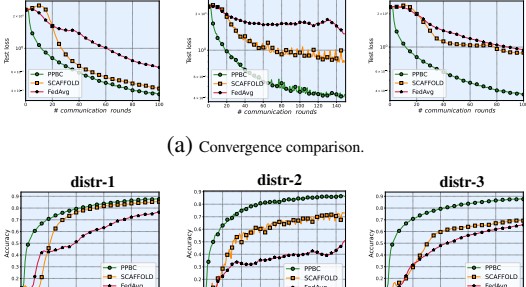

(a) Convergence comparison.

(b) Metrics comparison.

Figure 1: Performance comparison for PoC strategy with different data distributions.

**Trust-Score Sampling.**  The study by Xie et al. [2019] introduces the BANT, which implements

a trust-based sampling mechanism. This approach assigns dynamic trust scores to clients based on historical performance metrics. Thus, weight update rule can be described as:

1. The server assigns trust scores $\text{TS}_m^k$ to each client $m$ based on the alignment of their model updates with the performance on server-held ground truth data $\mathcal{V}$:

$$\text{TS}_m^k = \exp\left[-\frac{1}{|\mathcal{V}|}\sum_{\xi\in\mathcal{V}} f_m(x^k,\xi)\right].$$

2. The weights are updated with a probability proportional to the trust scores assigned to each client:

$$\pi^k = \left(\frac{\text{TS}_m^k}{\sum_{m'=1}^{M}\text{TS}_{m'}^k}\right)_{m=1}^{M}.$$

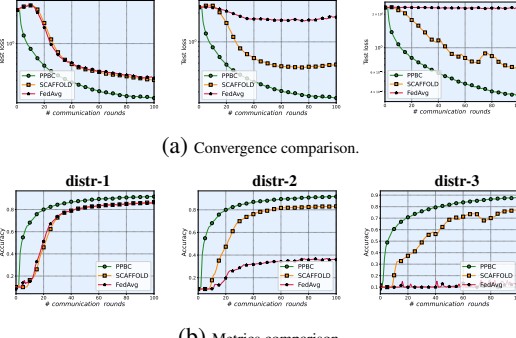

(a) Convergence comparison.

(b) Metrics comparison.

Figure 2: Performance comparison for BANT strategy with different data distributions.

**Importance Sampling.** Nguyen et al. [2020] introduced FOLB, a theoretically grounded client selection framework for federated learning that optimizes convergence by sampling clients proportionally to the expected utility of their local updates. The core selection mechanism operates as follows:

1. Each client is assigned an importance score $\text{IS}_m^k$ proportional to the inner product between its gradient $\nabla f_m(x^k,\xi_m^k)$ and the direction of the server model improvement (previous gradient $d^k$):

$$\text{IS}_m^k = \left|\left\langle\nabla f_m(x^k,\xi_m^k), d^k\right\rangle\right|.$$

2. The weights are updated with a probability proportional to the trust scores for each client:

$$\pi^k = \left(\frac{\text{IS}_m^k}{\sum_{m'=1}^{M}\text{IS}_{m'}^k}\right)_{m=1}^{M}.$$

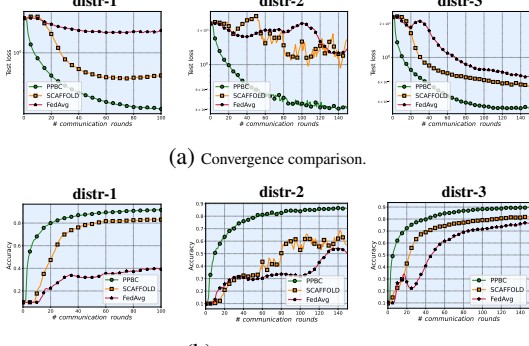

(a) Convergence comparison.

(b) Metrics comparison.

Figure 3: Performance comparison for FOLB strategy with different data distributions.

**Discussion.** Our experimental evaluation on the CIFAR-10 dataset using the RESNET18 architecture demonstrates a substantial performance gap between conventional approaches (FedAvg, SCAFFOLD) and Algorithm 1 (see Figures 1 2, and 3), providing strong empirical validation of our theoretical analysis. Notably, PPBC maintains consistent convergence rates and accuracy across all experimental configurations, with the observed performance variance remaining within 2% of theoretical predictions (see Figure 4). This robust empirical behavior confirms our key theoretical insight: PPBC's performance is strategy-agnostic, achieving stable convergence regardless of the underlying client selection mechanism.

We present additional experiments in Appendix B. We consider advanced client selection techniques that utilize the rule $\widetilde{\mathcal{R}}^{k,h}$, provide the results for PPBC+ (Algorithm 2), and demonstrate the outcomes for VIT training.

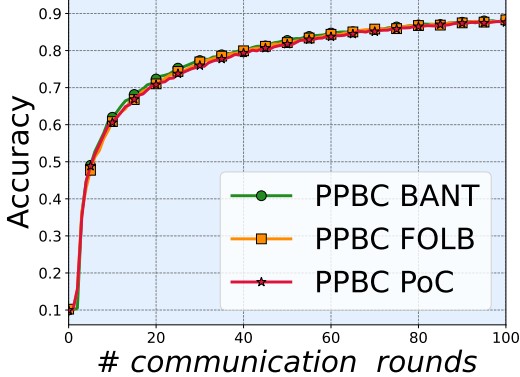

Figure 4: PPBC performace across all client-sampling strategies on the **distr-3**.

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
