# OpenReview forum: "A Robust Training Method for Federated Learning with Partial Participation"
_NeurIPS.cc/2025/Conference — Submitted to NeurIPS 2025_

### Official Review · Reviewer_NUFN · 2025-06-27

**Clarity:** 2
**Significance:** 4
**Originality:** 3
**Rating:** 5
**Confidence:** 1

**Summary:**

This paper introduces PPBC, a unified framework for federated learning that addresses the challenges of client weighting and partial participation. PPBC accumulates unused gradients locally during each round and incorporates them after global aggregation to improve training effectiveness. The method is compatible with a wide range of client sampling and weighting strategies and is robust to periodic client disconnections. Experimental results on both convolutional and transformer-based models demonstrate the effectiveness of PPBC in improving performance under partial participation scenarios.

**Questions:**

1) Convergence consistency across strategies: The experiments show that PPBC maintains similar convergence trends regardless of the underlying client sampling and weighting strategies applied.

* Could the authors elaborate on why convergence remains consistent across these strategies?

* Does this imply a generalizable property of PPBC that ensures stable convergence for any compatible strategy, including those not tested in the paper?

* A theoretical justification or an empirical investigation would greatly strengthen the contribution and clarify the mechanism behind this observation.

2) Lack of comparison with original strategies used in PPBC: The evaluation applies several existing strategies within the PPBC framework, but surprisingly, there is no direct comparison with the original versions of those strategies (outside PPBC).

* Why are such baseline comparisons omitted?

* Given that PPBC builds on these strategies, it would be valuable to show whether PPBC can match or even improve their performance under the same conditions.

Including these comparisons would improve the clarity of PPBC’s contributions and its fidelity in replicating diverse behaviors.

3) Choice of baselines (SCAFFOLD and FedAvg): The paper includes FedAvg and SCAFFOLD as baselines, but these are relatively dated.

* Could the authors justify the exclusion of more recent and competitive baselines, such as FedDyn, FedNova, or MOON, which are known to perform better in heterogeneous data settings?

* Including these comparisons would provide a stronger and more up-to-date empirical validation of PPBC.

**Ethical Concerns:**

["NO or VERY MINOR ethics concerns only"]

**Final Justification:**

The authors clarified the main concern of my review well, the concern about the missing comparison. I have a stronger faith in the work now. Therefore, I raised my rating.

**Limitations:**

yes

**Quality:**

3

**Strengths And Weaknesses:**

Strengths
+ Versatile and unifying framework: The proposed PPBC framework is highly versatile, supporting a wide range of client weighting and sampling strategies under a unified mechanism. This flexibility is a significant contribution, as it allows practitioners and researchers to explore and benchmark different federated learning strategies consistently within the same implementation.


Weakness
- Lack of baseline comparisons with original strategies: The paper evaluates PPBC by applying three existing client sampling/weighting strategies but does not compare PPBC's performance directly against the original implementations of those strategies. Since PPBC integrates these strategies, it is important to assess whether and how closely it replicates their expected training behavior and performance. Such comparisons are critical to verify the framework’s correctness and fidelity.

- Insufficient discussion of convergence consistency: One of the surprising results is that PPBC exhibits similar convergence behavior across different strategies. However, the paper does not provide sufficient explanation or analysis of this phenomenon. Understanding why PPBC converges robustly regardless of the underlying strategy would be important for interpreting its stability and generalization potential. A deeper discussion on this point would enhance the paper’s impact and clarity.

---

> ### Author Rebuttal · Authors · 2025-07-31
>
> Dear Reviewer NUFN!
>
> Thank you for evaluating our work! Below, we address your concerns.
>
> > The paper … does not compare PPBC's performance directly against the original implementations of those strategies.
>
> We respectfully disagree with the Reviewer's comment, as in all comparative experiments we indeed use the original strategies as described in the corresponding papers (see Section 4 in the main part and Section B in Appendix). Moreover, most of these methods are based on the FedAvg framework, whereas in our comparison we additionally include SCAFFOLD [4] (*ICML* 2020), which further extends and enriches the experimental evaluation.
>
> > However, the paper does not provide sufficient explanation or analysis of this phenomenon. Could the authors elaborate on why convergence remains consistent across these strategies?
>
> Thanks to Reviewer for their suggestion to improve our work! Classic partial participation approaches incorporate various strategies within the standard FedAvg framework, where performance varies because of bias in full gradient approximation $-$ with slower-accumulating bias strategies performing better. In our work, we observe a different phenomenon due to the use of gradient surrogates. Each active device accumulates local gradients when not transmitted to the server, allowing effective bias compensation at epoch completion. This leads to comparable performance across sampling strategies. We agree with Reviewer that this intuition must be discussed in our work. We will add more detailed explanations to prevent potential misunderstandings.
>
> > Does this imply a generalizable property of PPBC that ensures stable convergence for any compatible strategy, including those not tested in the paper?
>
> Thanks to Reviewer for their interesting question! The similar convergence behavior stems from our method's core design, indicating this is a systematic characteristic rather than coincidence. This property emerges from our bias compensation mechanism, which eliminates performance differences between sampling strategies.
>
> > A theoretical justification or an empirical investigation would greatly strengthen the contribution and clarify the mechanism behind this observation.
>
> We previously addressed this question informally, and our formal proof provides a rigorous explanation. Our gradient surrogates' specific formulation enables using full gradients in analysis (see Line 195). Consequently, our method's update behaves like classic parallel gradient descent while avoiding communication bottleneck.
>
> > The evaluation applies several existing strategies within the PPBC framework, but surprisingly, there is no direct comparison with the original versions of those strategies (outside PPBC). Why are such baseline comparisons omitted?
>
> We thank Reviewer for this thoughtful question and appreciate the opportunity to provide further clarification. The core contribution of our work lies in **enhancing the FedAvg framework** for FL through the local accumulation of gradient surrogates (as detailed in Section 1.4). Our primary experimental comparison focuses on the performance gap between **FedAvg and our proposed PPBC** framework under modern client selection protocols. We conduct pairwise comparisons between our approach and FedAvg, SCAFFOLD for each examined sampling strategy. We politely point out that Lines 267-271 states "To validate our theoretical findings, we conduct a systematic empirical study comparing three optimization approaches $-$ FedAvg, SCAFFOLD, and PPBC (Algorithm 1) — each integrated with the same client sampling technique. Crucially, we maintain a fixed strategy across all methods, deliberately decoupling the sampling mechanism from algorithmic innovations to focus specifically on its interaction with different optimization approach". To address possible concerns, we introduce additional state-of-the-art baselines, such as FedDyn [1] (*IEEE* 2024), Moon [2] (*IEEE* 2021), FedNova [3] (*IEEE* 2021) and baseline for scenarios with inactive clients F3AST [5] (*IEEE* 2022). We present new experiments under the most challenging non-iid setup, **distr-3**.  Namely, **distr-3** is a pathological data distribution $-$ clients have different amounts of data, and the distribution of sample sizes across clients is as follows:
>
>   | Client Number | Data Sample Proportion |
>   |-|-|
>   | 1 | 10.6% |
>   | 2 | 7.4%   |
>   | 3 | 12.0% |
>   | 4 | 11.4% |
>   | 5 | 8.8%   |
>   | 6 | 14.6% |
>   | 7 | 10.0% |
>   | 8 | 5.4%   |
>   | 9 | 10.2% |
>   | 10 | 9.2% |
>
> Within each client, class labels are sampled according to a Dirichlet distribution ($\alpha=0.5$), resulting in non-IID label distributions. We will include distributions details into our text. Firstly, we present the results for CIFAR10 classification problem across all client sampling strategies.
>
> PoC
> | Method | FedAvg | Scaffold | FedDyn | FedNova | Moon | PPBC  |
> |-|-|-|-|-|-|-|
> | Accuracy, % | 66.1 | 69.5 | 78.6 | 72.7 | 78.9 | 88.4 |
>
> BANT
> | Method | FedAvg | Scaffold | FedDyn | FedNova | Moon | PPBC  |
> |-|-|-|-|-|-|-|
> | Accuracy, % | 11.8 | 78.1 | 83.2 | 82.7 | 80.1 | 87.8 |
>
> FOLB
> | Method | FedAvg | Scaffold | FedDyn | FedNova | Moon | PPBC  |
> |-|-|-|-|-|-|-|
> | Accuracy, % | 76.2 | 82.1 | 80.9 | 83.4 | 79.2 | 89.6 |
>
> GNS
> | Method | FedAvg | Scaffold | FedDyn | FedNova | Moon | PPBC  |
> |-|-|-|-|-|-|-|
> | Accuracy, % | 76.1 | 81.7 | 84.5 | 82.2 | 79.2 | 89.8 |
>
> As demonstrated above, PPBC consistently outperforms all competing methods across various client participation strategies, achieving the highest accuracy in each setting. We further evaluate our PPBC+ algorithm in the context of fine-tuning FasterViT on Food101 dataset under varying levels of client participation probability $q_m$.
>
> $q_m = 1.0$
> | Method | FedAvg | F3AST | PPBC+ |
> |-|-|-|-|
> | Accuracy, % | 61.4 | 64.4 | 76.7 |
>
> $q_m = 0.7$
> | Method | FedAvg | F3AST | PPBC+ |
> |-|-|-|-|
> | Accuracy, % | 57.7 | 59.5 | 76.4 |
>
> $q_m = 0.5$
> | Method | FedAvg | F3AST | PPBC+ |
> |-|-|-|-|
> | Accuracy, % | 59.8 | 63.7 | 76.2 |
>
> $q_m = 0.3$
> | Method | FedAvg | F3AST | PPBC+ |
> |-|-|-|-|
> | Accuracy, % | 53.5 | 58.1 | 75.1 |
>
> The results clearly illustrate the consistent superiority of PPBC+ over existing approaches, especially under low client availability.
>
> These experiments further demonstrate that the consistent performance achieved by our framework is not coincidental, but rather a direct validation of our theoretical findings. The reproducible results across diverse experimental settings strongly support the theoretical foundations of our approach. We will include this study in our text.
>
> > Given that PPBC builds on these strategies, it would be valuable to show whether PPBC can match or even improve their performance under the same conditions
>
> We would like to emphasize that PPBC and PPBC+ not only achieve superior performance across all client selection rules but also demonstrate consistent convergence behavior throughout all evaluated selection strategies (see Section 4 in the main part and Section B in the Appendix) and additional experiments above.
>
> > Could the authors justify the exclusion of more recent and competitive baselines, such as FedDyn, FedNova, or MOON, which are known to perform better in heterogeneous data settings?
>
> See the reply above.
>
> In addition to the experiments requested by the reviewer we provide a comprehensive ablation study examining robustness to changes in hyperparameters $\theta$ and $p$. We begin our hyperparameter analysis by fixing $p = 0.2$ (which results in average epoch size equal to 5) and systematically varying the values of $\theta$ under the GNS client selection rule (see Lines 923-932). The results are summarized in the table below:
>
> | $\theta$ | Accuracy | Loss  |
> |-|-|-|
> | 0.05  | 0.88  | 0.35 |
> | 0.10  | 0.90  | 0.31 |
> | 0.15  | 0.93  | **0.21** |
> | 0.20  | 0.89  | 0.32 |
>
> We confirm our theoretical expectations: excessively small values of $\theta$ do not allow for effectively accounting for the clients'  history ($\theta = 0$ corresponds to FedAvg), while large values of $\theta$ disproportionately increases the contribution of gradient surrogates that become outdated after an epoch. However, there exists a wide interval within which the method do not lose much quality compared to optimal $\theta$. Next, we fix $\theta = 0.2$ and systematically vary the local epoch size for the same experimental setup. The results are presented in the table below:
>
> | $H^k$ | Accuracy | Loss  |
> |-|-|-|
> | 1 | 0.81 | 0.38 |
> | 3 | 0.91 | 0.23 |
> | 5 | 0.89 | 0.32 |
> | 7 | 0.82 | 0.39 |
>
> The results demonstrate that the optimal epoch size for $\theta = 0.15$ is 5 (see ablation study for $\theta$), while for $\theta = 0.2$, the optimal value decreases to 3.
>
> We appreciate the Reviewer's recognition of our method's strengths. We would also like to highlight that our PPBC+ extension provides robustness against periodic device disconnections, making it suitable for practical FL scenarios with unstable network connection. We remain available for any further discussion, and would be grateful if Reviewer could consider raising their score should they find our responses satisfactory.
>
> ---
> **References**
>
> [1] Bai, W. "Optimization of Federated Learning Algorithm for Non-lID Data...", **IEEE** 2024.
>
> [2] Li, Q. "Model-contrastive federated learning", **IEEE/CVF** 2021
>
> [3] Wang et al. "A novel framework for the analysis and design of heterogeneous federated learning", **IEEE** 2021
>
> [4] Karimireddy et al. "Scaffold: Stochastic controlled averaging for federated learning", **ICML** 2020
>
> [5] Ribero et al. "Federated learning under intermittent client availability and time-varying communication constraints", **IEEE** 2022

---

> > ### Comment · Reviewer_NUFN · 2025-08-05
> >
> > Thank you for your efforts in addressing my questions. I would like to request clarification regarding the following comment I made: "The paper … does not compare PPBC's performance directly against the original implementations of those strategies."
> >
> > Specifically, I was referring to the original implementations proposed by Cho et al. [2022], Xie et al. [2019], and Nguyen et al. [2022]. My intention was to understand whether the strategies implemented within PPBC closely follow those original versions. I feel that such a comparison would help demonstrate PPBC’s flexibility in supporting a range of strategies.
> >
> > Could you clarify whether you consider these comparisons necessary, or explain the reasoning behind not including them in the paper? Once I have a better understanding, I will be happy to reconsider my rating.

---

> ### Author Response · Authors · 2025-08-06
>
> Thank you for clarifying your question!
>
> First, we would like to emphasize that the methods mentioned in your comment can be viewed as special cases of our proposed framework, PPBC. For example, setting parameter $\theta = 0$ and client selection probabilities based on the clients’ loss values gives the original strategy presented in Cho et al. [2022]. Similarly, the original strategies presented in Xie et al. [2019] and Nguyen et al. [2022] can be obtained.
>
> In our experiments, we directly make a comparison with the original implementations of those sampling strategies. The legends of experiments Figures 1-3 indicate the frameworks rather than the strategies, since every plot compares the same original implementation of some strategy (**original baseline from Cho et al. [2022], Xie et al. [2019], or Nguyen et al. [2022] (FedAvg+strategy)** vs **Scaffold + strategy** vs **PPBC ($\theta > 0$) + strategy**), see Lines 267-271. This correspondence is formalized in Algorithm 1 of our paper, where the gradient surrogate term vanishes when $\theta = 0$, recovering the conventional update rule.
>
> Thus, our framework generalizes these approaches: by adjusting $\theta$ and the client weighting mechanism, one can recover the original methods, while also enabling a broader class of algorithms. This unifying perspective allows us to directly compare our method with the baselines under identical experimental conditions.
>
> In this context, there is an interest in ablation study on $\theta$, since there is a potential to achieve enhanced quality by varying this parameter. We have already provided such a study in our rebuttal, however, for Reviewer’s convenience we present it here. We conducted an ablation study with a fixed epoch size $H^k = 5$ and different $\theta$ values under the GNS client selection rule (see Lines 923-932):
> | $\theta$ | Accuracy | Loss |
> |-|-|-|
> | 0.00     | 0.76      | 0.72  |
> | 0.05     | 0.88      | 0.35  |
> | 0.10     | 0.90      | 0.31  |
> | 0.15     | 0.93      | 0.21  |
> | 0.20     | 0.89      | 0.32  |
>
> When $\theta=0$, the performance matches that of the traditional approach — as expected, since the algorithms are identical in this case. As $\theta$ increases, incorporating the gradient surrogate improves convergence, demonstrating clear gains over the baseline. However, if $\theta$ becomes too large, performance degrades due to overestimation of the surrogate, indicating the importance of balancing this correction term. Finally, we invite Reviewer to verify the implementation details in our publicly available code attached before the submission deadline.
>
> If you have any remaining questions, we are glad to continue the discussion.

---

> > ### Comment · Reviewer_NUFN · 2025-08-07
> >
> > Thank you for your response. This clears my concern. I raised my rating accordingly.

---

> ### Author Response · Authors · 2025-08-07
>
> Thanks to Reviewer for the discussion of our work!

---

### Official Review · Reviewer_dMGx · 2025-06-30

**Clarity:** 3
**Significance:** 2
**Originality:** 2
**Rating:** 5
**Confidence:** 3

**Summary:**

The paper proposes a method for partial client participation in the federate learning setup. The proposed method, Partial Participation with Bias Correction (PPBC) lets inactive federated-learning clients cache a gradient surrogate and leverages it during training. The authors provide theoretical analysis of PPBC convergence under different assumptions. Experiments on CIFAR-10 (ResNet-18) across three client-selection heuristics show faster loss reduction and higher accuracy than FedAvg and SCAFFOLD.

**Questions:**

***Questions***
1.  For the client selection schemes reviewed in the literature, do everyone of them assume inactive clients stay inactive through the entire epoch?
2. In algorithm 1 line 20, why are the surrogate gradients sent to the server at the end of all iterations? Why are the surrogate gradients not sent once between the iterations?
3. Referring back to line 115, how does algorithm 1 and the corresponding analysis, relaxes the requirement for boundedness of gradients or their difference.
4. In assumption 4 what does \eta signify?
5. In algorithm 1, why is there no step to account for clients receiving the updated model
6. From the algorithm and theory perspective, what are the core novel contributions
7. For the experiments presented in the main paper, how many clients were used.

**Ethical Concerns:**

["NO or VERY MINOR ethics concerns only"]

**Final Justification:**

I thank the authors for providing a detailed response to the questions that I had asked. In light of the additional results, further explanation and observing the discussion with other reviewers, I have decided to raise my score.

**Limitations:**

Yes

**Paper Formatting Concerns:**

None that were visible to me

**Quality:**

3

**Strengths And Weaknesses:**

***Strengths***
1) The paper is arranged in a very logical manner
2) The theoretical analysis covers different type of objectives and is quite elaborate
3) Extension of core algorithm (Algorithm 1) to account for unavailable device (Algorithm 2) is quite practical

***Weaknesses***

1. The introduction section is not easy to read. It talks about client weighting and client sampling, without mentioning clearly what the relevance is to the current research paper. In Line 116, it is mentioned that “there is still no flawless unified theory of partial participation”. However, it is not clear  why a  flawless unified theory is required and what are the impact of using FedAvg, which requires boundedness of gradients or their difference.

2. Using gradient surrogate seems to be a very important part of Algorithm 1, however, it is not explained how gradient surrogate is calculated and the intuition behind why it is helpful and makes the proposed algorithm perform well. In addition, the literature review does not mention much of prior research related to using gradient surrogates.

3. In section3.1 line 152, it’s mentioned that current methods yield poor results. However, it is not clear what are the results and why are they considered poor

4. No ablation study to understand the impact of gradient accumulation and impact of \theta

5. The idea of client sampling per section 1.2 is to reduce number of communications to the server, however line 20 in algorithm 1 will cause a communication bottleneck at the server.

---

> ### Author Rebuttal · Authors · 2025-07-31
>
> Dear Reviewer dMGx!
>
> Thank you for your time and assessment of our work! Further, we are responding to your claims and concerns.
>
> > The introduction section is not easy to read. It talks about client weighting and client sampling, without mentioning clearly what the relevance is to the current research paper.
>
> Thanks to Reviewer for valuable observation! We will expand our discussion of the work's objectives before reviewing client weighting and sampling strategies to improve the paper's readability. Before prior works review, we will first thoroughly examine the key challenges in federated learning and how these two approaches directly address them.
>
> > However, it is not clear why a flawless unified theory is required ….
>
> The community has accumulated numerous client weighting and sampling strategies [1], [2], [3]. In practice, this requires evaluating different approaches for specific tasks, increasing training costs. This creates demand for methods that can generalize sampling/weighting techniques. A flawless unified theory in federated learning is essential to enables a framework capable of incorporating arbitrary weighting and sampling strategies. We touched on this in Lines 97-100 but agree more detailed explanations would be helpful and will expand this section.
>
> >  … what are the impact of using FedAvg, which requires boundedness of gradients or their difference
>
> Indeed, vanilla FedAvg relies on unrealistic theoretical assumptions and has poor convergence guarantees. However, our work proposes an orthogonal approach not based on FedAvg, as explicitly stated in Lines 119-122. Note, our analysis operates under significantly weaker assumptions (see Section 2 for complete details).
>
> > … it is not explained how gradient surrogate is calculated and the intuition behind why it is helpful and makes the proposed algorithm perform well.
>
> We would like to kindly note that we provide substantial intuition about gradient surrogate in Lines 185-189. However, to address Reviewer's concerns, we expand the clarification. To compensate for client sampling bias, each device accumulates the difference between complete and transmitted gradient information. Simultaneously, since the goal is to learn average model, the surrogate incorporates correction for deviations between local model weights and uniform weighting. These two principles form the basis of gradient surrogate construction (Line 12, Algorithm 1).
>
>  > … the literature review does not mention much of prior research related to using gradient surrogates.
>
> Originally, this technique was developed for methods with compression (see Lines 166-172). However, to the best of our knowledge, we present the first use of gradient surrogates for setting with partial participation. We will dedicate separate paragraph in final version to surrogate technique discussion.
>
> > In section 3.1 line 152, it’s mentioned that current methods yield poor results. However, it is not clear what are the results and why are they considered poor.
>
> In Section 1.3, we thoroughly analyze theoretical guarantees of prior works that unify sampling strategies into a single framework, explicitly presenting convergence properties. Beyond relying on unrealistic assumptions, such as bounded stochastic gradients [4], [3], [5] or similarity between each local gradient and the global one [6], they establish convergence rates with improper asymptotic.
>
> > No ablation study to understand the impact of gradient accumulation and impact of $\theta$.
>
> Thanks to Reviewer for their valuable comment! We conducted ablation study with fixed epoch size $H^k = 5$ and different $\theta$ values under the GNS client selection rule (see Lines 923-932):
>
> |$\theta$|Accuracy|Loss|
> |-|-|-|
> |0.00|0.76|0.72|
> |0.05|0.88|0.35|
> |0.10|0.90|0.31|
> |0.15|0.93|0.21|
> |0.20|0.89|0.32|
>
> We confirm our theoretical expectations: small values of $\theta$ do not allow for effectively accounting for clients' history ($\theta=0$ corresponds to FedAvg), while large ones disproportionately increases contribution of gradient surrogates that become outdated after an epoch. However, there exists a wide interval within which the method maintains quality.
>
> > … line 20 in algorithm 1 will cause a communication bottleneck at the server.
>
> We acknowledge the Reviewer’s concern about the potential communication bottleneck. However, it occurs once per epoch. The fundamental principle of partial participation methods is precisely to reduce the frequency of full aggregation. This issue exists in all practical approaches including FedAvg and Scaffold [7]. Our algorithm's epoch size $H^k$ follows a geometric distribution with parameter $p$, enabling control via adjustment of $p$, while maintaining provable convergence for any $p$ (Theorems 1, 2, 5, 6). We support this claim by comprehensive ablation study: we fix $\theta = 0.2$ and vary $p$ for the same experimental setup as described above. Results are presented in the table:
>
> |$H^k$|Accuracy|Loss|
> |-|-|-|
> |1|0.81|0.38|
> |3|0.91|0.23|
> |5|0.89|0.32|
> |7|0.82|0.39|
>
> Combining the results from the previous question we demonstrate that the optimal epoch size for $\theta = 0.15$ is 5, while for $\theta = 0.2$, the optimal value decreases to 3. This finding is in complete agreement with theoretical expectations: smaller values of $\theta$ require fewer local epochs to achieve optimal convergence.
>
> > For the client selection schemes reviewed in the literature, do everyone of them assume inactive clients stay inactive through the entire epoch?
>
> While several schemes avoid communicating with poorly-connected clients (see Lines 90-100), to the best of our knowledge, existing literature does not consider this setup.
>
> > why are the surrogate gradients sent to the server at the end of all iterations? Why are the surrogate gradients not sent once between the iterations?
>
> In our method, the gradient surrogate is transmitted to the server after the epoch of size $H^k$ (see Lines 189-190, Algorithms 1 and 2). Exchanging surrogates after every iteration would violate the partial participation principles.
>
> > … how does algorithm 1 and the corresponding analysis, relaxes the requirement for boundedness of gradients
> or their difference.
>
> We thank Reviewer for their insightful question! Due to compensation of sampling bias via gradient surrogates, our scheme enables the use of full gradients in the convergence proofs while avoiding unrealistic assumptions prevalent in prior works (see proofs of Theorems 1, 2, 5, 6).
>
> > In assumption 4 what does \eta signify?
>
> We would kindly ask Reviewer to clarify their question. Assumption 4 does not use the notation $\eta$, but we believe they may be referring to $\alpha$ instead. This represents a standard assumption to avoid degenerate distribution of weights [8]. Theoretically, lacking such regularization would degrade the error bound by a factor of $M$, as degenerate distributions would imply communication with one device in the worst case.
>
> > In algorithm 1, why is there no step to account for clients receiving the updated model
>
> The literature frequently omits server-to-client communication steps to keep algorithms concise. To improve understanding of our work, we will add explicit steps in the algorithm listings showing the updated model being returned to devices.
>
> > From the algorithm and theory perspective, what are the core novel contributions
>
> We thank Reviewer for their insightful question about our work! As noted in our previous responses, the algorithmic innovation lies in proposing the first (as far as we know) biased partial participation scheme that is not built upon FedAvg. Theoretically, we develop a new error feedback framework, where corrections are applied epoch-wise rather than iteratively. The key theoretical challenges we overcome are detailed in Lines 196-200, 208-213 of our paper.
>
> > For the experiments presented in the main paper, how many clients were used.
>
> We thank Reviewer for the question! In our study, we employed 10 clients for both the  ResNet18 + CIFAR-10 setup and for the FasterViT fine-tuning on the Food101 dataset. This choice of client count was carefully selected to enable comprehensive evaluation across the diverse data distribution scenarios proposed in our work, while maintaining computational feasibility for thorough experimentation. If you want to see advanced details on our data distribution, please check tables in our answers to reviewers LJeY, dB2W, NUFN.
>
> > (PPBC) lets inactive federated-learning clients cache a gradient surrogate and leverages it during training.
>
> We appreciate Reviewer for highlighting this property of our method! However, we would like to note that the work extends PPBC to a modified version (see Algorithm 2 in Appendix A),  where some clients may lose connection with the server and consequently don't accumulate gradient surrogates for several iterations.
>
> > Experiments on CIFAR-10 (ResNet-18).
>
> We would like to kindly note that we also evaluate our methods on Food101 using FasterViT. This represents a substantially larger dataset with 101 classes and a more complex architecture containing 300M parameters. Reviewer can find these experimental results in Appendix B.
>
> If Reviewer has any remaining questions after our response, we would be happy to address them during the author-reviewer discussion. If all aspects have been clarified to the Reviewer's satisfaction, we would kindly ask them to reconsider their evaluation.
>
> ---
>
> **References**
>
> [1] Wang et al. Tackling the objective ... **NIPS 2020**
>
> [2] Gao et al. Feddc ... **IEEE 2022**
>
> [3] Cho et al. Towards understanding biased client ... **AISTATS 2022**
>
> [4] Li et al. On the Convergence of FedAvg ... **ICLR 2020**
>
> [5] Luo et al. Tackling system and statistical ... **IEEE2022**
>
> [6] Wang et al. A unified analysis of federated ... **NIPS 2022**
>
> [7] Karimireddy et al. Scaffold ... **ICML 2020**
>
> [8] Mehta et al. DRAGO ... **NIPS 2024**

---

> > ### Comment · Reviewer_dMGx · 2025-08-06
> >
> > Thank you for providing a response to the questions that I asked before. In regards to novelty, I think the proposed method, PPBC, does leverage some elements of FedAvg (aggregating gradients) but it seems that the primary novelty claimed is incorporation of error feedback (1,2), via gradient surrogates, to the FL context in a manner that does not allow all clients to account for the error at each iteration.
> > While this is a good technical contribution, I think this is a specific implementation detail and not a significant novel idea.
> > I'll maintain my score for the time being, but I will monitor the conversations with other reviewers and make updates as necessary.
> >
> >
> > [1]Sebastian U Stich and Sai Praneeth Karimireddy. The error-feedback framework: Sgd with delayed
> > gradients. Journal of Machine Learning Research, 21(237):1–36, 2020.
> >
> > [2]Peter Richtárik, Igor Sokolov, and Ilyas Fatkhullin. Ef21: A new, simpler, theoretically better,
> > and practically faster error feedback. Advances in Neural Information Processing Systems, 34:470
> > 4384–4396, 2021

---

> ### Author Response · Authors · 2025-08-07
>
> Thank you for your comment!
>
> 1. In your response, you mentioned the EF21 algorithm [1]. Indeed, this is a classic work that received an oral talk at **NeurIPS-21**. The update steps of this method are given by:
> $$g_m^{k+1} = g_m^k + C(\nabla f_m(x^k) - g_m^k),$$
> $$x^{k+1} = x^k - \gamma \frac{1}{M} \sum_{m=1}^M g_m^{k+1}.$$
> However, we would like to point out that EF21 is in some sense the simplification of the DIANA method [2]:
> $$h_m^{k+1} = h_m^k + \alpha C(\nabla f_m(x^k) - h_m^k),$$
> $$g_m^{k} = h_m^k + \frac{1}{M} \sum_{m=1}^M C(\nabla f_m(x^k) - h_m^k),$$
> $$x^{k+1} = x^k - \gamma \frac{1}{M} \sum_{m=1}^M g_m^{k}.$$
> Specifically, in addition to the surrogate $g^k$, DIANA exploits an extra surrogate $h^k$, which is updated recursively with a control parameter $\alpha$. When $\alpha = 1$, the surrogates $g^k$ and $h^k$ coincide, and DIANA reduces to EF21.  Nevertheless, the idea behind DIANA in some sense was not fully novel at the time of its publication in 2019 either. It was directly inspired by the variance-reduction scheme SEGA [3]:
> $$h^{k+1} = h^k + e_i^{\top}(\nabla f(x^k) - h^k)e_i,$$
> $$g^{k} = h^k + \alpha e_i^{\top}(\nabla f(x^k) - h^k)e_i,$$
> $$x^{k+1} = x^k - \gamma g^{k},$$
> where $e_i$ is the basis vector. In a certain sense, DIANA generalizes the idea of a coordinate compressor on a single device to arbitrary compressors on multiple devices. In turn, one can see that SEGA is a transfer of the SAGA technique [4] from finite-sum to coordinate setting. The update of SAGA is as follows:
> $$h_{m}^{k+1} = \nabla f_m(x^{k}),$$
> $$h_{i\neq m}^{k+1}=h_{i\neq m}^{k},$$
> $$g^k = \frac{1}{M}\sum_{i=1}^M h_{i}^{k} + (\nabla f_m(x^{k}) - h_{m}^{k}),$$
> $$x^{k+1} = x^k - \gamma g^k,$$
> where $m$ is the uniformly distributed variable in range $\overline{1, M}$.
> Thus, although gradient surrogates have been known in the optimization community for a long time, this area continues to actively evolve. We note that different variants of error feedback continue to appear at top-tier conferences [5], [6], [7], [8].
>
> 2. Surrogate-based algorithms mentioned by Reviewer are primarily focused on communication compression in distributed computing. In this context, there is a fundamental question: can we treat our approach as a method with compressed communications and straightforwardly apply ideas from the methods discussed above? The answer is no. The theory behind EF, EF21, and other existing error feedback methods relies on the assumption that the compressor is contractive:
> $$\mathbb{E}||\nabla f(x) - C(\nabla f(x))||^2 \leq \left(1 - \frac{1}{\beta}\right)||\nabla f(x)||^2.$$
> This condition is meaningful only for $1 \leq \beta < \infty$. Indeed, rates of traditional methods with compression scale as $\mathcal{O}(\beta)$. However, in our setting with partial participation, some devices do not send their gradients at all, which corresponds to compressing their outputs to zero i.e. $C(x)=0$. This means infinite power of compression ($\beta = \infty$), making compression operators no longer contractive and giving convergence guarantees as $\mathcal{O}(\infty)$. In fact, there is no convergence. This reasoning explains why the idea of adapting error feedback to the partial participation setting has not emerged in the community yet.
>
> 3. The argument in point 2 shows that the challenges we face are not merely technical. Indeed, it is unclear how to design an algorithm capable of handling non-contractive compression. Our novel insight is to interpret partial participation not as gradient compression, but as compression of a sampling probability vector. Thus, our work does not simply extend existing error feedback techniques, but rather shifts them into a new, orthogonal direction. In particular, we demonstrate that gradient surrogates have potential beyond the traditional gradient compression settings.
>
> ---
>
> **References**
>
> [1] Richtárik et al. "EF21: A new, simpler, theoretically better ...", **NIPS 2021**
>
> [2] Mishchenko et al. "Distributed Learning with Compressed ...", **arXiv 2019** (cited 272 times)
>
> [3] Hanzely et al. "SEGA: Variance reduction via gradient sketching", **NIPS 2018**
>
> [4] Defazio et al. "SAGA: A fast incremental gradient method ...", **NIPS 2014**
>
> [5] Condat et al. "EF-BV: A unified theory of error feedback ...", **NIPS 2022**
>
> [6] Li et al. "Analysis of error feedback in federated non-convex optimization ...", **ICML 2023**
>
> [7] Fatkhullin et al. "Momentum provably improves error feedback!", **NIPS 2023**
>
> [8] Islamov et al. "Safe-EF: Error Feedback for Non-smooth Constrained Optimization", **ICML 2025**

---

> > ### Comment · Reviewer_dMGx · 2025-08-08
> >
> > Thank you for your response.
> > I have looked at the discussion with other reviewers and decided to raise my score accordingly.

---

> > > ### Author Response · Authors · 2025-08-09
> > >
> > > Thank you for the fruitful discussion!

---

### Official Review · Reviewer_dB2W · 2025-06-30

**Clarity:** 3
**Significance:** 3
**Originality:** 3
**Rating:** 5
**Confidence:** 3

**Summary:**

This paper targets Federated Learning with partial client participation. The authors first proposed a method that allows devices to accumulate gradient surrogates locally, then let the server account for them after the full aggregation round. The proposed approach
supports various weighting and sampling client strategies, including biased ones. The authors also comprehensively analyzed method convergence in both strongly convex and non-convex cases.

**Questions:**

Please see the weakness and questions section.

**Ethical Concerns:**

["NO or VERY MINOR ethics concerns only"]

**Final Justification:**

Based on the author's replies during the rebuttal period, I believe they addressed most of my questions. So I will keep my score as 5.

**Limitations:**

Yes

**Quality:**

3

**Strengths And Weaknesses:**

Strengths:
1) The proposed framework supports a wide range of client weighting and sampling strategies, including biased ones.

2) The paper provides comprehensive convergence proofs for both non-convex and strongly convex settings, while most existing works focus primarily on convex assumptions.

3) It is novel and practical that Algorithm 2 explicitly considers device dropout, which frequently occurs in real-world federated learning scenarios.

Weaknesses and Questions:
1) Assumption 3 states that each gradient is similar to the full gradient. I have two questions regarding this assumption:
a) Does "each gradient" refer to the gradient on each client?
b) How can we ensure that the gradients remain sufficiently aligned, especially in highly non-iid scenarios? How robust is the method under poor gradient alignment caused by non-iid?

2) The method applies a stochastic operator to define the selection rule (e.g., zeroing out the weights of unavailable nodes). How does this operator handle newly joined or rejoining clients?

3) It is unclear how to determine the value of H^k (number of iterations per epoch) in Algorithm 1 (PPBC). How does this choice affect performance and convergence?

4) In PPBC, all inactive devices are required to accumulate gradient surrogates, which may introduce overhead. If a device has not been selected for an extended period, how is the growing memory or computational overhead handled?

5) Related to 3) and 4), if a device has not participated in the global aggregation for a long time, could its stale surrogate gradient distort or mislead the global update direction?

6) CIFAR-10 is a relatively small and balanced dataset that is not typically used for evaluating non-iid in federated learning. The method should also be tested on some non-iid benchmarks, such as LEAF.

---

> ### Author Rebuttal · Authors · 2025-07-31
>
> Dear Reviewer dB2W!
>
> Thank you for your time and high evaluation of our work! We would like to give comments regarding your concerns.
>
> > Assumption 3 states that each gradient is similar to the full gradient.
>
> We agree with Reviewer that providing convergence guarantees requires assuming than heterogeneity is bounded on average. Informally, this means that local gradients are indeed similar to the full gradient. However, this enables a scenario where some local gradients could be strongly deviating, but it is compensated by weak deviation of the rest ones, which is quite practical. Moreover, we would like to politely note that existing partial participation studies employ **much stricter** assumptions $-$ either bounded stochastic gradients [1] (*ICLR* 2020), [2] (*AISTATS* 2022), [3] (*IEEE* 2022) or similarity between individual local gradients and the global one [4] (*NeurIPS* 2022).
>
> > Does "each gradient" refer to the gradient on each client?
>
> Yes, "each gradient" refers to the local gradient of each client computed on their own data. We will add this clarification to our text to ensure reader comprehension.
>
> > How can we ensure that the gradients remain sufficiently aligned, especially in highly non-iid scenarios?
>
> We acknowledge this assumption may appear restrictive, but it is actually mild. In highly non-IID settings, the values of $\delta_1$ and $\delta_2$ grow, leading to increase of convergence bounds (see the exact dependence in Corollaries 1, 2, 3, 4). We also would like to point out that our scheme converges even with poor alignment of the gradients.
>
> > How robust is the method under poor gradient alignment caused by non-iid?
>
> We thank Reviewer for their interesting question! Our methods outperform the competitors in homogeneous scenario, and its final metrics do not drastically decline with non-iid data (see **distr-2** and **distr-3** in Figures 1-3 for ResNet-18 and Figure 6 for FasterViT). A detailed description of the non-iid scenarios examined in our study can be found in Lines 276-283. We agree that the description of the data distributions used may not have been sufficiently clear. Therefore, we provide below a detailed summary of the data distribution characteristics for each experimental setup.
>
> **Distr-1 (Homogeneous):** Homogeneous data distribution — each client has the same number of data samples, and class labels are uniformly distributed across clients.
>   *Example (CIFAR-10):* Each client has 500 training samples per class, resulting in 5,000 samples per client in total.
>
> **Distr-2 (Heterogeneous):** Heterogeneous data distribution — each client has the same total number of samples, but class labels are distributed in a non-IID manner.
>   *Example (CIFAR-10):* We split the classes into two groups (e.g., classes 0–4 and classes 5–9), and clients are divided into two corresponding groups. Clients in each group only receive data from the classes assigned to their group. Additionally, the number of samples per class varies across clients.
>
> **Distr-3 (Pathological):** Pathological data distribution — clients have different amounts of data. The distribution of sample sizes across clients is as follows:
>
>   | Client Number | Data Sample Proportion |
>   |-|-|
>   | 1 | 10.6% |
>   | 2 | 7.4%   |
>   | 3 | 12.0% |
>   | 4 | 11.4% |
>   | 5 | 8.8%   |
>   | 6 | 14.6% |
>   | 7 | 10.0% |
>   | 8 | 5.4%   |
>   | 9 | 10.2% |
>   | 10 | 9.2% |
>
> Within each client, class labels are sampled according to a Dirichlet distribution ($\alpha = 0.5$), resulting in non-IID label distributions. We are willing to add this description into our text.
>
> > How does this operator handle newly joined or rejoining clients?
>
> We employ the stochastic operator as a formalization of partial participation. Our theoretical analysis does not require the operator to remain fixed during training (see Corollaries 6, 8, 10, 12), including with regard to the number of clients. This specifically means our framework covers the device join and rejoin scenarios mentioned by Reviewer.
>
> > If a device has not been selected for an extended period, how is the growing memory or computational overhead handled?
>
> Our algorithm maintains efficient memory usage on devices: at each iteration, we overwrite the new surrogate gradient value without retaining the old one (see Line 12 in Algorithm 1). Thus, the memory overhead boils down to storing a copy of the global model on each device and poses no greater computational challenge than in classical federated approaches.
>
> > if a device has not participated in the global aggregation for a long time, could its stale surrogate gradient distort or mislead the global update direction?
>
> The duration without global aggregation is determined by epoch size $H^k$, a geometrically distributed random variable with parameter $p$ (Line 8 in Algorithms 1, 2). Our analysis imposes no restrictions on $p$, with theoretical convergence being dependent on this hyperparameter (see details in Theorems 1, 2, 5, 6). Notably, our algorithm includes the hyperparameter $\theta$ that compensates for deviation in the gradient surrogate's direction. From parameter tuning perspective, smaller values of $p$ require smaller $\theta$. We provide a comprehensive ablation study examining $\theta$ and $p$ configurations.
>
> We begin our hyperparameter analysis by fixing $p = 0.2$ (which results in average epoch size equal to 5) and systematically varying the values of $\theta$ under the GNS client selection rule (see Lines 923-932). The results are summarized in the table below:
>
> | $\theta$ | Accuracy | Loss  |
> |-|-|-|
> | 0.05  | 0.88  | 0.35 |
> | 0.10  | 0.90  | 0.31 |
> | 0.15  | 0.93  | **0.21** |
> | 0.20  | 0.89  | 0.32 |
>
> We confirm our theoretical expectations: excessively small values of $\theta$ do not allow for effectively accounting for the clients'  history ($\theta = 0$ corresponds to FedAvg), while large values of $\theta$ disproportionately increases the contribution of gradient surrogates that become outdated after an epoch. However, there exists a wide interval within which the method do not lose much quality compared to optimal $\theta$. Next, we fix $\theta = 0.2$ and systematically vary the local epoch size for the same experimental setup. The results are presented in the table below:
>
> | $H^k$ | Accuracy | Loss  |
> |-|-|-|
> | 1 | 0.81 | 0.38 |
> | 3 | 0.91 | 0.23 |
> | 5 | 0.89 | 0.32 |
> | 7 | 0.82 | 0.39 |
>
> The results demonstrate that the optimal epoch size for $\theta = 0.15$ is 5 (see ablation study for $\theta$), while for $\theta = 0.2$, the optimal value decreases to 3. This finding is in complete agreement with theoretical expectations: bigger values of $\theta$ require fewer number of local steps to achieve optimal convergence. This empirical validation further strengthens the theoretical foundations of our approach.
>
> > CIFAR-10 is a relatively small and balanced dataset that is not typically used for evaluating non-iid in federated learning. The method should also be tested on some non-iid benchmarks, such as LEAF.
>
> We acknowledge Reviewer’s point that CIFAR-10 is relatively small for modern applications. However, our work also includes tests on Food101 classification with FasterViT (see Figures 6, 8). It is a 101K images dataset with 101 classes. Additionally, we would like to emphasize that three types of data distributions are used in the comparative experiments (see Lines 276–283), ranging from homogeneous to non-iid. We provide above a detailed summary of the data distribution characteristics for each experimental setup.
>
> If Reviewer has any remaining questions after our response, we would be happy to continue the discussion.
>
> ---
>
> **References**
>
> [1] Li, X., Huang, K., Yang, W., Wang, S., & Zhang, Z. On the Convergence of FedAvg on Non-IID Data. In **International Conference on Learning Representations**.
>
> [2] Yae Jee Cho, Jianyu Wang, and Gauri Joshi. Towards understanding biased client selection in federated learning. In **International Conference on Artificial Intelligence and Statistics**, pages 10351–10375. PMLR, 2022.
>
> [3] Bing Luo, Wenli Xiao, Shiqiang Wang, Jianwei Huang, and Leandros Tassiulas. Tackling system and statistical heterogeneity for federated learning with adaptive client sampling. In **IEEE** INFOCOM 2022-IEEE conference on computer communications, pages 1739–1748. IEEE, 2022.
>
> [4] Shiqiang Wang and Mingyue Ji. A unified analysis of federated learning with arbitrary client participation. **Advances in Neural Information Processing Systems**, 35:19124–19137, 2022.

---

> > ### Comment · Reviewer_dB2W · 2025-08-03
> >
> > Thank you for replying. I do not have further questions.

---

> > > ### Author Response · Authors · 2025-08-07
> > >
> > > Thanks to Reviewer for the response to our rebuttal and for evaluating our work!

---

### Official Review · Reviewer_LJeY · 2025-07-03

**Clarity:** 3
**Significance:** 3
**Originality:** 3
**Rating:** 4
**Confidence:** 2

**Summary:**

Prior works on partial participation analysis are typically based on the standard FedAvg algorithm, which does not account for inactive clients. This paper introduces the PPBC, a framework designed for scenarios involving inactive clients. The PPBC framework is general enough to encompass various client weighting and sampling strategies. The authors provide a theoretical analysis of PPBC's convergence rate for both convex and non-convex objectives.

**Questions:**

Please refer to the weaknesses.

**Ethical Concerns:**

["NO or VERY MINOR ethics concerns only"]

**Final Justification:**

The authors have provided sufficient responses to the weaknesses and questions I initially raised. I believe the issues I perceived as weaknesses have been mostly resolved, which in turn provides sufficient justification for the contributions claimed in the paper.
However, it is still difficult to readily grasp the necessity and significance of the methodology proposed in the paper.

**Limitations:**

yes

**Paper Formatting Concerns:**

The citation format is inappropriate.

**Quality:**

3

**Strengths And Weaknesses:**

### **Strengths**

S1. The proposed framework is comprehensive, as it covers a wide range of previously studied client weighting and sampling approaches.

S2. The authors provide a rigorous theoretical analysis of the PPBC's convergence.

S3. Experiments on CIFAR-10 and Food-101 with ResNet-18 and FasterViT confirm the method's effectiveness.

### **Weaknesses**

W1. The paper is difficult to follow; its overall structure and writing require refinement.

W2. The authors claim that leveraging information from inactive clients is one of the key contributions. However, the experiments presented in the main paper are exclusively for the PPBC variant. To support the claim, results for PPBC+ must be included and emphasized in the main paper.

W3. The set of baselines appears insufficient. A comparison with other methods designed for settings with inactive clients is necessary.

W4. In Section 3.3, the line numbers in Algorithm 2 do not match those in Appendix A.

---

> ### Author Rebuttal · Authors · 2025-07-31
>
> Dear Reviewer LJeY!
>
> Thank you for your time and assessment of our work! Further, we are responding to your concerns and claims.
>
> > The paper is difficult to follow; its overall structure and writing require refinement.
>
> We thank Reviewer for their feedback on our paper's structure! During the reviewing process, we performed multiple proofreads of the text and made every effort to address any points that could potentially lead to misunderstanding. In the current version, we thoroughly examine the key challenges in federated learning and how weighting and sampling directly address them before delving into a prior works review. Furthermore, we dedicate a separate paragraph to reviewing gradient surrogates in compression methods and expand on how this approach could be adapted for partial participation scenarios.
>
> > However, the experiments presented in the main paper are exclusively for the PPBC variant. To support the claim, results for PPBC+ must be included and emphasized in the main paper.
>
> We would like to highlight that the results for PPBC+ are provided in Appendix attached to the submission on OpenReview (this is pointed out at Lines 337-338, Section 4). However, we recognize the importance of scenarios with inactive clients in federated learning. We will move the PPBC+ empirical validation to the main part.
>
> > The set of baselines appears insufficient. A comparison with other methods designed for settings with inactive clients is necessary.
>
> We thank Reviewer for their valuable feedback! We fully agree that incorporating additional baselines for settings with inactive clients is crucial to strengthen the paper’s empirical validation. We include F3AST [4] (*IEEE*, 2022) to address your concern. We evaluate our PPBC+ algorithm in the context of fine-tuning FasterViT on Food101 dataset under varying levels of client participation probability $q_m$.
>
> $q_m = 1.0$
> | Method | FedAvg | F3AST | PPBC+ |
> |-|-|-|-|
> | Accuracy, % | 61.4 | 64.4 | 76.7 |
>
> $q_m = 0.7$
> | Method | FedAvg | F3AST | PPBC+ |
> |-|-|-|-|
> | Accuracy, % | 57.7 | 59.5 | 76.4 |
>
> $q_m = 0.5$
> | Method | FedAvg | F3AST | PPBC+ |
> |-|-|-|-|
> | Accuracy, % | 59.8 | 63.7 | 76.2 |
>
> $q_m = 0.3$
> | Method | FedAvg | F3AST | PPBC+ |
> |-|-|-|-|
> | Accuracy, % | 53.5 | 58.1 | 75.1 |
>
> These results clearly illustrate the consistent superiority of PPBC+ over existing approaches, especially under low client availability. In particular, PPBC+ outperforms F3AST by a large margin across all settings, and it remains stable even when only 30% of clients participate on average. We will include this empirical result to the main part of our paper.
>
> Moreover, we have significantly expanded our experimental evaluation by including several representative FL baselines: FedDyn [1] (*IEEE*, 2024), Moon [2] (*IEEE*, 2021), and FedNova [3] (*IEEE*, 2021). All experiments are presented under the most challenging non-iid setup, **distr-3** (Lines 276-278). Namely, **distr-3** is pathological data distribution -- clients have different amounts of data, and the distribution of sample sizes across clients is as follows:
>
>   | Client Number | Data Sample Proportion |
>   |-|-|
>   | 1 | 10.6% |
>   | 2 | 7.4%   |
>   | 3 | 12.0% |
>   | 4 | 11.4% |
>   | 5 | 8.8%   |
>   | 6 | 14.6% |
>   | 7 | 10.0% |
>   | 8 | 5.4%   |
>   | 9 | 10.2% |
>   | 10 | 9.2% |
>
> Within each client, class labels are sampled according to a Dirichlet distribution ($\alpha = 0.5$), resulting in non-IID label distributions. We will include distributions details into our text. We present the results for CIFAR10 classification problem across all client sampling strategies (see descriptions in Section 4 and Appendix B).
>
> PoC
> | Method | FedAvg | Scaffold | FedDyn | FedNova | Moon | PPBC  |
> |-|-|-|-|-|-|-|
> | Accuracy, % | 66.1 | 69.5 | 78.6 | 72.7 | 78.9 | 88.4 |
>
> BANT
> | Method | FedAvg | Scaffold | FedDyn | FedNova | Moon | PPBC  |
> |-|-|-|-|-|-|-|
> | Accuracy, % | 11.8 | 78.1 | 83.2 | 82.7 | 80.1 | 87.8 |
>
> FOLB
> | Method | FedAvg | Scaffold | FedDyn | FedNova | Moon | PPBC  |
> |-|-|-|-|-|-|-|
> | Accuracy, % | 76.2 | 82.1 | 80.9 | 83.4 | 79.2 | 89.6 |
>
> GNS
> | Method | FedAvg | Scaffold | FedDyn | FedNova | Moon | PPBC  |
> |-|-|-|-|-|-|-|
> | Accuracy, % | 76.1 | 81.7 | 84.5 | 82.2 | 79.2 | 89.8 |
>
> As demonstrated above, PPBC consistently outperforms all competing methods across various client participation strategies, achieving the highest accuracy in each setting.
>
> > In Section 3.3, the line numbers in Algorithm 2 do not match those in Appendix A.
>
> Thanks, typo!
>
> > Prior works on partial participation analysis are typically based on the standard FedAvg algorithm, which does not account for inactive clients.
>
> We would like to additionally note that FedAvg-based approaches have certain limitations. Their analysis requires restrictive assumptions, such as bounded stochastic gradients [6] (*ICLR*, 2020), [5] (*AISTATS*, 2022), [7] (*IEEE*, 2022), or uniformly bounded deviation of each local gradient from the full one [8] (*NeurIPS*, 2022). Furthermore, even under these assumptions, they demonstrate suboptimal convergence rate asymptotics.
>
> > The authors provide a theoretical analysis of PPBC's convergence rate for both convex and non-convex objectives.
>
> Thanks to the Reviewer for highlighting this part of our work! However, we would like to emphasize that we not only analyze the convergence of the proposed method, but also derive correct estimates under weaker assumptions than those found in existing literature on the subject.
>
> We would like to note that most of the Reviewer's concerns relate to the paper's formatting, such as correcting identified typos and moving some figures from the Appendix to the main part. These points can be easily addressed. Regarding baselines, we agree that the tested methods fall under data-based sampling approaches. We have included comparisons with strategies specifically designed for scenarios involving inactive clients. If we have addressed all questions satisfactorily, we would kindly ask Reviewer to reconsider their evaluation.
>
> ---
>
> **References**
>
> [1] Bai, W. (2024, November). Optimization of Federated Learning Algorithm for Non-lID Data: Improvements to the FedDyn Algorithm. In 2024 International Conference on Computing, Robotics and System Sciences (ICRSS) (pp. 277-283). **IEEE**.
>
> [2] Li, Q., He, B., & Song, D. (2021). Model-contrastive federated learning. In Proceedings of the **IEEE/CVF** conference on computer vision and pattern recognition (pp. 10713-10722).
>
> [3] Wang, J., Liu, Q., Liang, H., Joshi, G., & Poor, H. V. (2021). A novel framework for the analysis and design of heterogeneous federated learning. **IEEE** Transactions on Signal Processing, 69, 5234-5249.
>
> [4] Ribero, M., Vikalo, H., & De Veciana, G. (2022). Federated learning under intermittent client availability and time-varying communication constraints. **IEEE** Journal of Selected Topics in Signal Processing, 17(1), 98-111.
>
> [5] Yae Jee Cho, Jianyu Wang, and Gauri Joshi. Towards understanding biased client selection in federated learning. In **International Conference on Artificial Intelligence and Statistics**, pages 10351–10375. PMLR, 2022.
>
> [6] Li, X., Huang, K., Yang, W., Wang, S., & Zhang, Z. On the Convergence of FedAvg on Non-IID Data. In **International Conference on Learning Representations**.
>
> [7] Bing Luo, Wenli Xiao, Shiqiang Wang, Jianwei Huang, and Leandros Tassiulas. Tackling system and statistical heterogeneity for federated learning with adaptive client sampling. In **IEEE** INFOCOM 2022-IEEE conference on computer communications, pages 1739–1748. IEEE, 2022.
>
> [8] Shiqiang Wang and Mingyue Ji. A unified analysis of federated learning with arbitrary client participation. **Advances in Neural Information Processing Systems**, 35:19124–19137, 2022.

---

> > ### Comment · Reviewer_LJeY · 2025-08-06
> >
> > Thank you for the clarification. I have no further questions.
> >
> > I will re-evaluate the paper, reflecting the points that PPBC+ shows better performance than F3AST designed for inactive clients and that the experimental results for PPBC+ will be included in the main paper.

---

> > > ### Author Response · Authors · 2025-08-07
> > >
> > > Thanks to Reviewer for engaging in the discussion and for valuable suggestions on how to improve the experimental section of our work!

---

### Decision · Program_Chairs · 2025-09-17

**Decision:**

Reject

**Comment:**

The reviewers initially mentioned a number of weaknesses and questions, ranging from the paper being difficult to follow, insufficient experiments, and lack of clarity regarding various parts of the proposed method (different reviewers had questions about different parts).
At the same time, there was a rather unanimous agreement that the paper proposes a comprehensive framework, very well-grounded in theory, that unifies a number of existing methods and practical use cases.

With respect to the initial submission, I tend to agree especially with the criticism concerning limited evaluation - only one model and dataset was included in the main paper (plus one more pair in the Appendix, but this one is not compared to any extra baseline other than FedAvg), although the paper does try to make up for it with its theoretical analysis. More importantly, however, the authors seem to have been able to provide important missing experiments during the rebuttal period, together with clarifying the parts of the paper that reviewers had asked for. As a result, all reviewers now unanimously recommend acceptance, and given the combination of the 3 mentioned things: 1) extensive theoretical analysis, 2) (new) experiments evidencing improvements in model training, and 3) detailed explanations from the authors to the reviewers’ questions; I do not see a reason to contradict this assessment. In particular, the only last outstanding shortcoming seems to be related to the fact that the necessity of the proposed method can be somewhat unclear. Still, in the light of all the above, I do not think that this would suffice for rejecting the paper at this point (it was only mentioned by one of the reviewers).

Having said that, I hope the authors incorporate necessary changes in the camera-ready version, as it is somewhat clear the reviewers needed additional convincing and clarifications.

===

As recently advised by legal counsel, the NeurIPS Foundation is unable to provide services, including the publication of academic articles, involving the technology sector of the Russian Federation’s economy under a sanction order laid out in Executive Order (E.O.) 14024.

Based upon a manual review of institutions, one or more of the authors listed on this paper submission has ties to organizations listed in E.O. 14024. As a result this paper has been identified as falling under this requirement and therefore must not be accepted under E.O. 14024.

This decision may be revisited if all authors on this paper can provide proof that their institutions are not listed under E.O. 14024 to the NeurIPS PC and legal teams before October 2, 2025. Final decisions will be communicated soon after October 2nd. Appeals may be directed to pc2025@neurips.cc.